# *Helicobacter pylori* Biofilm Confers Antibiotic Tolerance in Part via A Protein-Dependent Mechanism

**DOI:** 10.3390/antibiotics9060355

**Published:** 2020-06-24

**Authors:** Skander Hathroubi, Julia Zerebinski, Aaron Clarke, Karen M. Ottemann

**Affiliations:** 1Department of Microbiology and Environmental Toxicology, University of California, Santa Cruz, CA 95064, USA; 2Institüt für Biologie/Mikrobiologie, Humboldt-Universität zu Berlin, 10115 Berlin, Germany; jzerebin81@yahoo.com (J.Z.); abclarke@ucsc.edu (A.C.)

**Keywords:** *Helicobacter pylori*, biofilms, antibiotic tolerance, extracellular proteins, proteinase K

## Abstract

*Helicobacter pylori*, a WHO class I carcinogen, is one of the most successful human pathogens colonizing the stomach of over 4.4 billion of the world’s population. Antibiotic therapy represents the best solution but poor response rates have hampered the elimination of *H. pylori*. A growing body of evidence suggests that *H. pylori* forms biofilms, but the role of this growth mode in infection remains elusive. Here, we demonstrate that *H. pylori* cells within a biofilm are tolerant to multiple antibiotics in a manner that depends partially on extracellular proteins. Biofilm-forming cells were tolerant to multiple antibiotics that target distinct pathways, including amoxicillin, clarithromycin, and tetracycline. Furthermore, this tolerance was significantly dampened following proteinase K treatment. These data suggest that *H. pylori* adapts its phenotype during biofilm growth resulting in decreased antibiotic susceptibility but this tolerance can be partially ameliorated by extracellular protease treatment.

## 1. Introduction

*Helicobacter pylori* chronically colonizes the stomachs of approximately half of the global population [1]. Although the prevalence differs between nations, infection rates remain high in developing countries (80–90%). Infections are often asymptomatic, although chronic infections can lead to gastritis, peptic ulcers, and may even develop to mucosa-associated lymphoid tissue lymphoma and gastric cancer. Gastric cancer kills over 700,000 people per year worldwide and has an estimated economic burden of over 6 billion dollars in the United State of America [1,2]. 

*Helicobacter pylori* infections are treatable through antibiotic therapy, which can reduce the incidence of cancers and ulcerations [3]. Despite the availability of antibiotic interventions, the successful eradication of *H. pylori* remains challenging [4]. Typical therapy consists of a 10–14-day course of proton pump inhibitor (PPI) and two to three antibiotics (clarithromycin, metronidazole, amoxicillin, levofloxacin, or tetracycline). Although combination therapies are effective in the majority of patients, ~15–25% of cases require additional rounds of treatment to resolve the infection [5,6,7]. 

There are two leading theories as to why *H. pylori* infections are difficult to cure: antibiotic resistance and antibiotic tolerance. Resistance of commonly used antibiotics to treat *H. pylori* infections has significantly increased in recent years, and more than doubling during the past 20 years in Europe [8,9]. In some countries such as Austria, Hungary, and Portugal, resistance to clarithromycin has increased substantially (36.6%, 33.3%, and 31.5%, respectively), and has limited the viability of this drug for routine treatment [8]. In the United States, the prevalence of clarithromycin resistance exceeds 40% in Northwestern hospitals [10]. To address the alarming rate at which these microbes are developing resistance, the World Health Organization (WHO) recently classified clarithromycin-resistant *H. pylori* as a high priority for antibiotic research and the development of alternative therapies [10]. 

Conditional antibiotic tolerance also certainly plays a role. *H. pylori* are rarely resistant to all antibiotics used [6], and therapies are dramatically improved by the addition of PPI [11]. Although the PPI mechanism is not fully understood, one role for PPIs seems to be to promote *H. pylori* growth and thus render it more susceptible to antibiotic treatment [12]. 

The in vivo growth state of *H. pylori* is not well understood, but a common state of growth associated with antibiotic tolerance is formation of biofilm [13,14,15,16]. Biofilm-forming cells of many microbes can be 10 to 10,000 times more tolerant to antibiotics than their planktonic counterparts [13,17,18]. Recent studies have suggested that biofilm-grown *H. pylori* are tolerant to the commonly used antibiotics clarithromycin, amoxicillin, and metronidazole, and these conditions lead *H. pylori* to express high levels of antibiotic-associated efflux pumps [19,20]. These studies examined a limited set of *H. pylori* clinical strains, so it is not yet clear whether this tolerance mechanism is widespread. This finding suggests, however, that biofilm formation may contribute to *H. pylori* persistence and treatment failure.

Biofilms are a major global challenge to modern medicine and their role in antibiotic tolerance and resistance is no longer questioned. In *H. pylori*, however, little has been done to explore the role and mechanisms involved in biofilm-mediated antibiotic tolerance. In the present study, we evaluated the antibiotic susceptibility of the *H. pylori* model strain G27, which is a robust biofilm-forming strain and tested the hypothesis that extracellular proteins might play a role in antibiotic tolerance. 

## 2. Results

### 2.1. H. Pylori G27 Biofilm Cells are Antibiotic Tolerant

We first asked whether *H. pylori* G27 biofilms were antibiotic tolerant by determining susceptibility of biofilm and planktonic cells to three clinically relevant antibiotics: the macrolide clarithromycin, the β-lactam amoxicillin, and tetracycline. *H. pylori* G27 biofilms were formed for 48 h, left untreated or treated with antibiotic for another 24 h, and then biofilm biomass was evaluated using the crystal violet assay as described in the Methods. Prior to biofilm treatment, we determined the *H. pylori* G27 minimum inhibitory concentrations (MIC) for each antibiotic using the ETEST strip method. We found that the MICs for clarithromycin was 0.094 mg/L, amoxicillin was 0.125 mg/L, and tetracycline was 0.75 mg/L. We then used these amounts to guide treatment of *H. pylori* G27 biofilm. Generally, antibiotic treatment had some effect but did not fully eliminate the biofilms. Amoxicillin treatment did not clearly diminish the biofilm biomass at up to 128 times the MIC (Figure 1), whereas tetracycline caused no effect at up to 64 times the MIC and a significant 30% decrease at 128 times the MIC (Figure 1). Clarithromycin was somewhat more effective, causing a 60% decrease in biomass at even four times the MIC, but still leaving 30% of the biomass at even the highest dose (Figure 1). Overall, these data suggest that biofilm *H. pylori* are not eliminated by high antibiotic dose. 

We next assessed the viability of biofilm and planktonic cells after exposure to the antibiotics. This analysis was accomplished by plating the samples to analyze live cells by colony-forming units (CFU), compared to untreated controls to determine the percent killing (Figure 2A). As expected, planktonic cells were completely killed by exposure to 128 times MIC levels of each antibiotic, resulting in no viable cells (Figure 2A). Conversely, cells grown as biofilms tolerated exposure to amoxicillin or tetracycline up to 128 times MIC, with only 12.5% or 7.5% of biofilm cell populations killed, respectively (Figure 2A). Clarithromycin was more able to kill biofilm cells with exposure at 128 times MIC, eliminating approximately half the population (Figure 2A). These results suggest that a substantial number of *H. pylori* biofilm cells survive high level antibiotic exposure.

We further evaluated the impact of clarithromycin on biofilms using confocal microscopy (Figure 2B,C). As shown previously [21], untreated *H. pylori* biofilms contain both live and dead cells (Figure 2B,C). After high dose treatment, we observed a large number of dead cells. We analyzed the distribution of dead and surviving cells after clarithromycin exposure, but did not observe any noticeable differences between cells at the top or bottom layers (Figure 2C). This result suggests that treatment with up to 128 times of the MIC of clarithromycin increases dead cells however, dead population are homogeneously distributed within the biofilm (Figure 2C). Additionally, we determined that around 50% of the population survived the clarithromycin treatment (Figure 2B,C), similar to the finding with CFU analysis. Together, these data show that *H. pylori* G27 biofilm cells show high tolerance towards antimicrobials, and combined with published data [19,20], suggests that antibiotic tolerance may be a general property of *H. pylori* biofilm cells. 

### 2.2. Biofilm-Associated Proteins Restrict Clarithromycin Effects

Biofilms can constitute physical and chemical diffusion barriers to antimicrobial penetration [13,17]. *H. pylori* is known to use a matrix composed mainly of proteins [21,22,23,24]. We thus tested whether *H. pylori*’s extracellular proteins limit antibiotic action. The biofilms were treated with proteinase K, which was shown previously to disperse preformed *H. pylori* G27 biofilms [22]. We first determined the biofilm proteinase K dose response, to identify doses that would largely leave preformed biofilms intact, with the idea that the biofilm structure would be partially destabilized and allow diffusion of antibiotics. Doses between 0.5 and 5 μg/mL of proteinase K left the biofilm intact (Figure 3A) and did not affect the viability of strain G27 (data not shown). Combining proteinase K (0.5 μg/mL) with clarithromycin at 128 times the MIC resulted in a significantly higher biofilm eradication compared to treatment with clarithromycin alone (Figure 3B). This outcome suggests that some target of proteinase K, such as the extracellular matrix or outer membrane proteins, limits clarithromycin exposure. These combined treatments did not completely eradicate *H. pylori* biofilms, so there may be other properties associated with *H. pylori* biofilm growth that contribute to clarithromycin resistance, such as low metabolism or overexpression of efflux pumps [19,21,25]. Our data thus show that *H. pylori* biofilms are antibiotic tolerant, and this phenotype is partially dependent on extracellular proteins.

## 3. Discussion

We report here that *H. pylori* biofilm formation leads to antibiotic tolerance in a manner that is partially dependent on extracellular proteins. Biofilm cells are highly tolerant to amoxicillin and tetracycline and moderately tolerant to clarithromycin. This clarithromycin tolerance can be alleviated by low-level treatment with proteinase K, suggesting the antibiotic tolerance state of the biofilms is at least partially reversible one. 

Despite recent interest, *H. pylori* biofilms are still relatively understudied in general, with little information about their direct and indirect roles in antibiotic tolerance and resistance. The reason for this lack of attention is probably multifactorial. First, *H. pylori* is a highly fastidious microorganism that requires complex growth media containing blood or serum, low-oxygen condition, and long incubations. Growing *H. pylori* as a biofilm makes things even more difficult, since it requires additional specific conditions and longer incubation time. It was recently shown that *H. pylori* SS1, a strain model for studying disease pathogenesis in murine experiment, requires low levels of serum to form a robust biofilm. Low levels of serum are not optimal for planktonic growth but it seems to trigger a stress or signaling that drastically increases the biofilm formation and thus sessile mode of growth [21]. As shown here and previously, these conditions are not necessary for *H. pylori* G27 to form biofilms. To date, there is the lack of consensus about the best strain with which to study *H. pylori* biofilms and also about the ideal methodology. Indeed, several approaches have been developed to investigate *H. pylori* biofilms across laboratories including glass frits [26], glass slides [27,28], and borosilicate glass bottles [29] for the study of biofilms formed at the air–liquid interface, and plastic tissue culture plates [21,22] or biotic surface such as MDCK epithelial cells [27] for biofilms formed at the solid–liquid interface. These variations in strains and approaches make the comparisons challenging. 

Here, we used *H. pylori* strain G27, which has benefits because it is commonly available and well characterized with reliable growth and biofilm formation [21,22]. We used the 96-well plate model, which allows the formation of biofilm at the solid–liquid and air–liquid interfaces. Within this biofilm, DNA and proteins were the most important components of the matrix described to be unevenly distributed [22]. G27 biofilms have been previously shown to be inhibited by high dose of proteinase K (25–200 µg/mL) [22]. 

Proteinase K is a relatively nonspecific protease that does not breach the *H. pylori* outer membrane at doses below 400 µg/mL [30]. We, thus, predict that our 5 µg/mL proteinase K dose largely targets proteins outside the outer membrane such as those secreted in biofilm matrix or associated with the outer membrane. Within *H. pylori*, matrix proteins have been suggested to be the main and the primary structural component, based on confocal microscopy and enzymatic assays [22]. However, the matrix does not have a substantial appearance in electron microscopy [21], raising the question of how well it would actually limit small molecule diffusion. Indeed, recent evidence suggests that even thick biofilm matrices do not substantially slow diffusion of small molecules such as antibiotics [31,32,33]. Another possibility for the proteinase K targets is outer membrane proteins [34]. These proteins could directly limit antibiotic penetration by coating the outer membrane, act as adhesins to connect *H. pylori* cells into structures that block antibiotic exposure of inner biofilm layers, or could be a component of tripartite efflux pumps [35,36,37,38]. Outer membrane proteins constitute a diverse group that includes porins, which fulfill multiple roles, including diffusion of small molecules such as antibiotics including β-lactams and fluoroquinolones [36,37,38,39]. In a study of *H. pylori* outer membrane proteins, the Hop and Hom family proteins were susceptible to proteolytic cleavage, whereas the Hor and Hof family proteins are resistant to proteinase K treatment [34]. Hop proteins are designated as porins [40], although their influence on antibiotic susceptibility in *H. pylori* remains largely unexplored.

In other microbes, there are known biofilm-promoting surface proteins that are sensitive to proteinase K. Specifically, the biofilm-associated protein (Bap) are cell-anchored proteins that promote biofilm development in both Gram-negative and Gram-positive bacteria [41]. In *Staphylococcus aureus*, 2 μg/mL proteinase K significantly inhibited biofilm development of Bap-positive strains but did not affect Bap-deficient mutant strains [42]. Orthologs to Bap exist in other bacterial species [43], including LapA in *Pseudomonas fluorescens*. LapA is described as an cell-anchored adhesin that is required for attachment and biofilm formation and is necessary to adhere to abiotic surfaces [44]. LapA is found in the G27 genome and shares a 37% domain identity to *P. fluorescens* LapA, providing a potential target for proteinase K cleavage in *H. pylori*. Thus, it seems likely that proteinase K acts on *H. pylori*-anchored surface proteins, but their identity and function remain to be investigated. 

Several groups have documented increased expression of efflux pumps in *H. pylori* biofilm cells. Since some of these efflux pumps span the outer membrane, these might be rendered nonfunctional by proteinase K. Previous work showed that *H. pylori* clinical strains, TK1049 and TK1402, became antibiotic tolerant in the biofilm state to amoxicillin, metronidazole, and clarithromycin [19,20]. These results generally agreed well with those presented here, although there were some differences. The most substantial was that TK1049 and TK1402 biofilms did not become completely tolerant to amoxicillin, with amoxicillin treatment resulting in ~40% reduction in biofilm mass as compared to no reduction with the strain used here, G27 [20]. Clarithromycin treatment gave similar results with TK1402 and G27; in that, the antibiotic partially removed and killed biofilm cells [19]. Overall, these studies all converge on the idea that biofilm growth renders *H. pylori* antibiotic tolerant, but the degree of tolerance appears to vary between strain backgrounds and the antibiotic used. 

Biofilm cells are also known to have altered physiology in addition to expressing surface structures [45]. Recent work showed that *H. pylori* biofilms have indicators of slowed growth. Transcriptome analysis revealed that key metabolic genes of the TCA cycle were downregulated in biofilm cells compared to their planktonic counterparts [21]. Biofilms are well known to generate heterogeneous gradients of nutrients and oxygen with microniches that drastically mitigate cell growth and metabolic states [45]. Growth and metabolic states are crucial determinants for antibiotic susceptibility [46], however, in *H. pylori*, the biofilm slow-growth impact on antibiotic susceptibilities has yet to be determined. 

Tolerance is well known to play a crucial role in the survival of the bacterial population [14]. Antibiotic tolerance and persistence also represent a reservoir for bacteria to acquire mutations [14]. In a recent study using an evolution experiment and mathematical modeling, tolerance preceded and enhanced the evolution of resistance when cells were subjected to intermittent antibiotic exposure [47]. Knowing that *H. pylori* treatments require long-term regimen that include several antibiotics, and that we are facing alarming levels of antibiotic resistance, tolerance and its impact on therapies in *H. pylori* must be taken into consideration in the future. 

In summary, our study suggests that *H. pylori* biofilm growth creates *H. pylori* cells that are significantly antibiotic tolerant. Our data suggest that this phenotype arises from a combination of extracellular proteins as well as other mechanism that has yet to be fully characterized. It will be exciting to learn in more detail how antibiotic tolerance is created, how it affects antibiotic resistance, and how it can be prevented to promote more effective *H. pylori* treatments. 

## 4. Materials and Methods

### 4.1. Bacterial Strain and Growth Conditions

This study employed *H. pylori* G27 strain, generously provided by Nina Salama [48]. *H. pylori* was grown on Columbia Horse Blood Agar (CHBA) (Difco), containing: 0.2% β-cyclodextrin, 10 μg/mL vancomycin, 5 μg/mL cefsulodin, 2.5 U/mL polymyxin B, 5 μg/mL trimethoprim, and 8 μg/mL amphotericin (all chemicals are from Thermo Fisher or Gold Biotech), or Brucella Broth (Difco) containing 10% heat-inactivated fetal bovine serum (BB10) (Gibco/BRL). Culture were grown under microaerobic conditions (10% CO_2_, 5% O_2_, and 85% N_2_) at 37 °C. For planktonic growth, constant shaking (180 rpm) was used. 

### 4.2. Biofilm Formation and Crystal Violet Assays

For biofilm formation, *H. pylori* strains were grown overnight with shaking in BB10, diluted to an OD_600_ of 0.15 with fresh BB10 media, and 200 µL of the bacterial suspension was added to the wells of a sterile 96-well polystyrene microtiter plates (Costar, 3596). Experiments were performed in biological and technical triplicates. Following 3 days of static incubation under microaerobic and static conditions, the culture medium was removed by aspiration and the wells were washed twice with phosphate-buffered saline (PBS). Wells were then filled with 200 μL of crystal violet (0.1%, wt/vol), and plates were incubated for 2 min at room temperature. The crystal violet solution was removed by aspiration, the wells were washed twice with PBS, and then, air dried for 20 min at room temperature. To visualize biofilms, 200 μL of 70% ethanol (vol/vol) was added to each well, and the absorbance was measured at 590.

### 4.3. Antibiotic Minimum Inhibitory Concentration (MIC) Determination 

The antimicrobial susceptibilities for amoxicillin, clarithromycin, and tetracycline were determined using ETEST strips (bioMérieux). BB10 cultures of *H. pylori* were adjusted overnight to an OD_600_ of 0.5, and 50 µL of culture was spread onto a CHBA plate. ETEST strips were aseptically placed onto each plate and incubated for 72 h. MICs were identified as the point at which bacterial growth intersected the strip. The experiments were conducted in triplicate for each antibiotic. 

### 4.4. Planktonic and Biofilm Antimicrobial Susceptibility

To determine the antibiotic susceptibility of biofilm cells, they were cultured as described above in 96-well plates. After 48 h of incubation, the supernatant was removed by aspiration and 200 μL of BB10 media containing the desired antibiotic concentration was added to each well. Wells containing media without antibiotic supplementation were used as positive controls, and wells containing antibiotic supplementations without culture were used as negative control. Plates were incubated for 24 h before biofilm-containing wells were washed, stained, and quantified with crystal violet staining as described above. Alternatively, biofilms with and without exposure to clarithromycin were washed in 2X PBS, resuspended in PBS, and serially diluted to count colony-forming units (CFUs). For planktonic growth assays, *H. pylori* cultures were grown for 18–72 h and then exposed to the desired antibiotic concentration for 24 h. Cultures were then plated for enumeration by CFU counting. 

### 4.5. Biofilm Dispersion Assay

Proteinase K treatment of established biofilms was performed as described previously [21,22]. Briefly, biofilms were grown as above. After 48 h of incubation, supernatants were replaced with either fresh media containing either proteinase K alone (200–0.05 μg/mL, Sigma-Aldrich) or fresh media containing proteinase K (200–0.05 μg/mL) and clarithromycin (MIC or 128× MIC). For control wells, supernatants were replaced with fresh media. Wells were treated for 24 h under the desired conditions, and biofilms were quantified using crystal violet staining as described above. 

### 4.6. Confocal Laser Scanning Microscopy

Biofilms of *H. pylori* G27 WT were prepared as described above using BB10 media. To use confocal laser scanning microscopy (CLSM), μ-Slide 8-well glass bottom chamber slides (ibidi, Germany) were used in place of 96-well microtiter plates. Growth and treatment were the same for other antibiotic exposure experiments. After 3 days, biofilms were stained using a FilmTracer™ FM^®^1–43 (Invitrogen) or FilmTracer™ LIVE/DEAD Biofilm Viability Kit (Invitrogen) according to the manufacturer’s instructions. Stained biofilms were visualized by CLSM with an LSM 5 Pascal Laser Scanning Microscope (Zeiss), and images were acquired using Imaris software (Bitplane). Biomass analysis of biofilm was carried out using FM^®^1–43 stained z-stack images (0.15 μm thickness) obtained by CLSM from randomly selected areas. The biomass of biofilms was determined using COMSTAT [49]. 

### 4.7. Statistical Analysis

Biofilm data were analyzed statistically using GraphPad Prism software (version 8, GraphPad Software Inc., San Diego, CA) by application of Wilcoxon–Mann–Whitney test or one-way ANOVA with Bartlett’s test. *p* < 0.05 or < 0.01 were considered to reflect statistical significance.

## Figures and Tables

**Figure 1 antibiotics-09-00355-f001:**
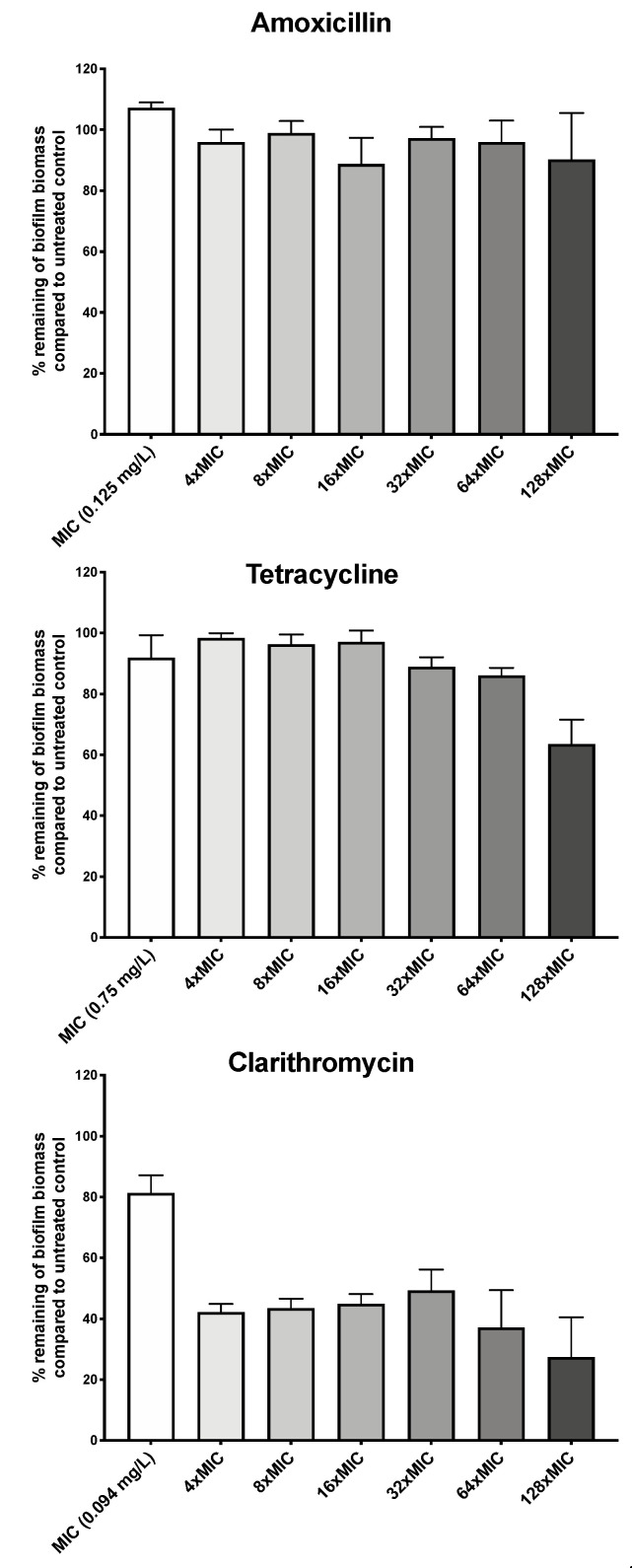
*H. pylori* biofilm growth is associated with high level tolerance towards amoxicillin, tetracycline, and clarithromycin. The percentage of remaining *H. pylori* G27 biofilm biomass after exposure to different concentration of amoxicillin, tetracycline, or clarithromycin compared to untreated control. Biofilms were grown for 48 h and then, exposed to antibiotics at the indicated concentrations for 24 h. Control/untreated biofilms were exposed to fresh media without antibiotic for the same time period. Biofilm were quantified using crystal violet. The percent biomass remaining was determined relative to the untreated control. Experiments were performed three independent times with at least 8 technical replicates for each. Error bars represent standard errors of the mean.

**Figure 2 antibiotics-09-00355-f002:**
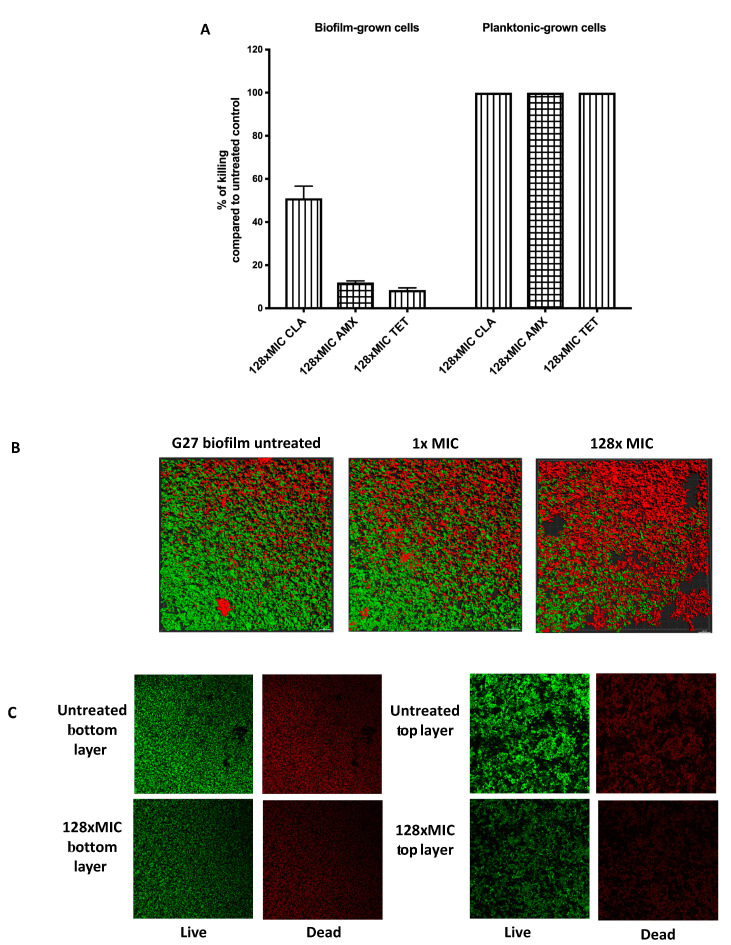
Viability of *H. pylori* biofilm and planktonic cells after antibiotic exposure. (**A**) *H. pylori* G27 were grown in biofilm or planktonic conditions for 48 h and then treated or not treated with either clarithromycin (CLR), amoxicillin (AMX), and tetracycline (TET) for an additional 24 h, as described in Figure 1. After treatment, samples were plated to determine colony-forming units (CFU) and compared to the untreated control, to determine the % killing. (**B**) Live and dead biomasses in *H. pylori* G27 biofilm treated or not treated with clarithromycin. Biofilms were visualized by confocal laser scanning microscopy (CLSM) with LIVE/DEAD viability strain. Viable cells exhibit green fluorescence, whereas dead and damaged cells exhibit red fluorescence. These images are three-dimensional (3D) reconstructions from an average of 19 image stacks assembled using Imaris software. (**C**) Images were taken from the bottom and top layers of *H. pylori* G27 biofilm treated or not treated with 128× MIC of clarithromycin.

**Figure 3 antibiotics-09-00355-f003:**
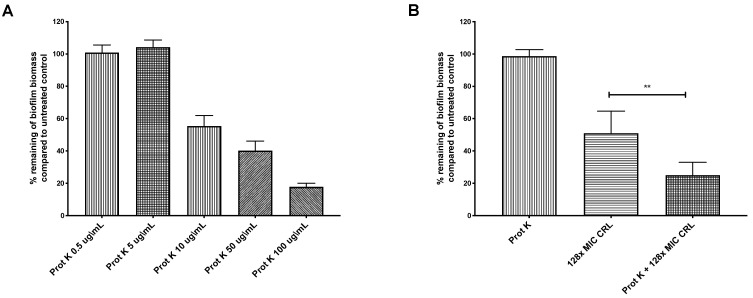
Proteinase K treatment sensitizes *H. pylori* G27 biofilms to clarithromycin. *H. pylori* G27 was grown as a biofilm for 48 h (as done in Figure 1) before exposure to varying doses of proteinase K (Prot K) ± clarithromycin (CRL) for 24 h. The biofilm biomass was determined by crystal violet staining, and compared to untreated controls (fresh media without antibiotic and/or proteinase K). (**A**) Proteinase K (Prot K) dose response. (**B**) Proteinase K at 5 µg/mL (Prot K), clarithromycin (CRL), or both treatments. Experiments were performed three independent times with at least eight technical replicates for each. Error bars represent standard errors of the mean. Statistical analyses were performed using ANOVA (**, *p* < 0.01).

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
