# Peer review of "Helicobacter pylori Biofilm Confers Antibiotic Tolerance in Part via A Protein-Dependent Mechanism"

_antibiotics, 2020, doi:10.3390/antibiotics9060355_

Round 1

Reviewer 1 Report

Manuscript No. antibiotics-830327 and entitled “Helicobacter pylori biofilm confer antibiotic tolerance in part via a protein-dependent mechanismprovides a new dataset, and these results may be of interest to the readership of the Antibiotics journal. Moreover, it may be well cited in the future. Although the work (its results) are quite modest.

Overall, the experimental design and data analysis are appropriate, and the introduction and discussion are consistent enough. However, please broaden (expand) the discussion, which is quite short, even as a communication. Nevertheless, in my opinion, this work should make a good contribution to the literature.

My comments

1) The name of the microorganism we write the full name when it begins the sentence.

2) Please do not start paragraphs after each other by the name “Helicobacter pylori”. Please "smooth" the text and use word substitutes or the manuscript should be checked by Native speaker.

3) Pg 8 Ln 213 “0.2% -cyclodextrin” Why this dash?

4) Statistical analysis – please add licenses for the program.

Author Response

Overall, the experimental design and data analysis are appropriate, and the introduction and discussion are consistent enough. However, please broaden (expand) the discussion, which is quite short, even as a communication. Nevertheless, in my opinion, this work should make a good contribution to the literature.

Response to reviewer 1:

We thank the reviewer for all these comments.

In 2017, WHO listed antibiotic-resistant Helicobacter pylori as an important threat to human health. However, there remain several substantial gaps in our understanding of why H. pylori is difficult to cure with antibiotics, including information about the biofilm mode of growth, specifically its impact on antibiotic tolerance and resistance as well as the mechanisms behind. There have been few studies on H. pylori biofilm and even less on the antibiotic tolerance during this mode of growth; specifically, only 2 previous papers addressed H. pylori responses to antibiotics when in a biofilm (both referred to in the text). We have modified the manuscript to highlight the current lack of information and research about H. pylori biofilm, to make this point clearer. We anticipate this study will pave the way for future research to combat biofilm-related antibiotic tolerance and better understand the mechanism behind it.

My comments

  • The name of the microorganism we write the full name when it begins the sentence.

We agree with the reviewer and we have addressed this concern throughout the text.

  • Please do not start paragraphs after each other by the name “Helicobacter pylori”. Please "smooth" the text and use word substitutes or the manuscript should be checked by Native speaker.

We thank the reviewer for this point and we have edited the text to address some minor errors, and it was carefully read and edited by a native English speaker.

  • Pg 8 Ln 213 “0.2% -cyclodextrin” Why this dash?

We thank the reviewer for this point. There were a β missing in the sentence and it was added and corrected for β-cyclodextrin (line 246).

  • Statistical analysis – please add licenses for the program.

We clarified that we use Prism 8 for macOS.

Reviewer 2 Report

Skander et al. submitted a short manuscript for publication as communication paper in the MDPI journal Antibiotics. The manuscript deals with the investigation of resistance/tolerance properties of H. pylori – a pathogen classified as high priority by the WHO – when cultivated as planktonic cells or under biofilm inducing conditions. The authors show that biofilm forming cells show antibiotic resistance/tolerance towards three different antibiotics. They could also show that proteinase treatment of biofilms diminishes the extent of resistance/tolerance towards one of the three antibiotics of interest.

Major comments:

Line 73: Please provide the original data of the MIC assay as part of figure 1

Figure 2: Please provide the single planes of the stack of Fig. 2B 128x MIC. It appears, that only the surface cells are dead, whereas the buried cells are alive. This must be clarified.

Lines 171 to 175: You give examples for cell anchored proteins involved in the antibiotics-resistance response of S. aureus. From my personal point of view the current manuscript has rather a descriptive style and its message is not as relevant as it might be. At least, the authors should screen for ortholog genes between S. aureus and H. pylori (both genomes are sequenced) to check if the proteins they mention are present in H. pylori as well. Probably, the authors already did so. Because the next step would be to prepare corresponding mutants for these proteins to verify their impact on resistance… By this experiment one could also test if there are also relevant extracellular matrix proteins addressed by the proteinase K.

Lines 210 to 219: This part of the Mat and Met does not fit to the content of the paper. Please correct. You did not show results for mutants or a table 1, for example.

Minor comments:

Line 13: With “Treatment” you mean treatment by antibiotics?

Line 15: change to …suggests…

Line 15: change to …biofilms…

Line 48: change to…seems to be to promote…

Line 77: Because you did not evaluate the significance of the data using a corresponding assay you should not use the term “significantly” here but rather “clearly”

Lines 77, 78, and 80: change to …Fig. 1… (instead of Figure. 1)

Figure 1 and Figure 3: Please correct the y-axis of each diagram > remaining and biomass

Line 115: change to …biofilm cells show high tolerance towards antimicrobials, …

Line 130: change to …biofilm eradication compared to…

Line 143: insert a space between 5 and µg/ml

Line 149: change to …partially dependent on proteins of the outer membrane or the extracellular matrix. The tolerance is not dependent on the enzyme itself!

Lines 154, 155, and 156: change to …outer membrane…

Line 168: Please provide the explanation (outer membrane protein) for OMP here

Line 179: insert a comma after amoxicillin

Line 180: delete “this” before “those”

Line 187: change to …Biofilm cells…

Line 190: change to …Biofilms…

Line 197: delete the comma after experiment

Line 202: change to …H. pylori cells that…

Line 213: “0.2%     -cyclodextrin” is not clear to me

Line 217: insert a space between 37 and °C

Line 222: Please explain composition of BB10

Line 234: Are the overnight cultures from static or shake cultivation?

Line 236: What medium was used with the plates?

Line 246: remove the comma between “and” and “quantified”

Line 263: What kind of mutants do you mean? (remove)

Author Response

Skander et al. submitted a short manuscript for publication as a communication paper in the MDPI journal Antibiotics. The manuscript deals with the investigation of resistance/tolerance properties of H. pylori – a pathogen classified as high priority by the WHO – when cultivated as planktonic cells or under biofilm inducing conditions. The authors show that biofilm forming cells show antibiotic resistance/tolerance towards three different antibiotics. They could also show that proteinase treatment of biofilms diminishes the extent of resistance/tolerance towards one of the three antibiotics of interest.

Major comments:

Line 73: Please provide the original data of the MIC assay as part of figure 1

We agree with the reviewer and we have modified figure 1 so MIC values are presented.

Figure 2: Please provide the single planes of the stack of Fig. 2B 128x MIC. It appears that only the surface cells are dead, whereas the buried cells are alive. This must be clarified.

We agree with this point and we added new images of the top and bottom layers of biofilm treated or not with ATBs. We modified the footnote and added some text to explain it and added new sentences to address this concern (lines 94-102).

Lines 171 to 175: You give examples for cell anchored proteins involved in the antibiotics-resistance response of S. aureus. From my personal point of view the current manuscript has rather a descriptive style and its message is not as relevant as it might be. At least, the authors should screen for ortholog genes between S. aureus and H. pylori (both genomes are sequenced) to check if the proteins they mention are present in H. pylori. Probably, the authors already did so. Because the next step would be to prepare corresponding mutants for these proteins to verify their impact on resistance… By this experiment one could also test if there are also relevant extracellular matrix proteins addressed by the proteinase K.

We added the example of Bap and LapA to emphasize the role of some cell anchored proteins in biofilms formed by other species. However, in H. pylori the role of Hop and Hom in biofilm formation and antibiotic tolerance has yet to be determined. As mentioned by the reviewer the next step will be to test some of the mutants affected in genes encoding for Hop and Hom family proteins and verify their impact on antibiotic tolerance and resistance. The text was modified accordingly (lines 192-202 and 205-209).

Lines 210 to 219: This part of the Mat and Met does not fit to the content of the paper. Please correct. You did not show results for mutants or a table 1, for example.

We thank the reviewer for pointing out this wrong sentence and we have fixed it now (line 245-251).

Minor comments:

Line 13: With “Treatment” you mean treatment by antibiotics?

We changed treatment to antibiotic therapy (line 13).

Line 15: change to …suggests…

We changed it accordingly.

Line 15: change to …biofilms…

We changed it accordingly.

Line 48: change to…seems to be to promote…

Changed to seems to promote (line 49).

Line 77: Because you did not evaluate the significance of the data using a corresponding assay you should not use the term “significantly” here but rather “clearly”

Changed to clearly (line 78).

Lines 77, 78, and 80: change to …Fig. 1… (instead of Figure. 1)

Changed accordingly.

Figure 1 and Figure 3: Please correct the y-axis of each diagram > remaining and biomass

We thank you for pointing out this typo and we fixed both figures 1 and 3.

Line 115: change to …biofilm cells show high tolerance towards antimicrobials, …

Changed accordingly.

Line 130: change to …biofilm eradication compared to…

Changed accordingly.

Line 143: insert a space between 5 and µg/ml

Changed accordingly.

Line 149: change to …partially dependent on proteins of the outer membrane or the extracellular matrix. The tolerance is not dependent on the enzyme itself!

Changed accordingly.

Lines 154, 155, and 156: change to …outer membrane…

Changed accordingly.

Line 168: Please provide the explanation (outer membrane protein) for OMP here

Added accordingly.

Line 179: insert a comma after amoxicillin

Corrected.

Line 180: delete “this” before “those”

Changed accordingly.

Line 187: change to …Biofilm cells…

Changed accordingly.

Line 190: change to …Biofilms…

Changed accordingly.

Line 197: delete the comma after experiment

Changed accordingly.

Line 202: change to …H. pylori cells that…

Changed accordingly.

Line 213: “0.2%     -cyclodextrin” is not clear to me

Changed to β-cyclodextrin.

Line 217: insert a space between 37 and °C

Changed accordingly.

Line 222: Please explain composition of BB10

Changed accordingly.

Line 234: Are the overnight cultures from static or shake cultivation?

Overnight cultures are under shaking (lines 250-251). biofilm cultures however were always cultivated in static condition (line 257). We have modified the text to point out these two points.  

Line 236: What medium was used with the plates?

We used 200 uL of bacterila suspension (BB10) for each well with at least technical and biological triplicates (Line 255).

Line 246: remove the comma between “and” and “quantified”

Changed accordingly.

Line 263: What kind of mutants do you mean? (remove)

We thank the reviewer for pointing out this typo we deleted this sentence.

Round 2

Reviewer 2 Report

The authors have addressed my concerns in a satisfying manner.